# Investigating the Performance of Oxford Nanopore Long-Read Sequencing with Respect to Illumina Microarrays and Short-Read Sequencing

**DOI:** 10.3390/ijms26104492

**Published:** 2025-05-08

**Authors:** Renato Santos, Hyunah Lee, Alexander Williams, Anastasia Baffour-Kyei, Sang-Hyuck Lee, Claire Troakes, Ammar Al-Chalabi, Gerome Breen, Alfredo Iacoangeli

**Affiliations:** 1Department of Biostatistics & Health Informatics, Institute of Psychiatry Psychology & Neuroscience, King’s College London, 16 De Crespigny Park, London SE5 8AB, UK; renato.santos@kcl.ac.uk; 2Social Genetic and Developmental Psychiatry Centre, Institute of Psychiatry Psychology & Neuroscience, King’s College London, 16 De Crespigny Park, London SE5 8AB, UK; hyunah.1.lee@kcl.ac.uk (H.L.); anastasia.baffour-kyei@kcl.ac.uk (A.B.-K.); sang_hyuck.lee@kcl.ac.uk (S.-H.L.); gerome.breen@kcl.ac.uk (G.B.); 3Department of Basic and Clinical Neuroscience, Institute of Psychiatry Psychology & Neuroscience, King’s College London, 5 Cutcombe Rd, London SE5 9RX, UK; claire.troakes@kcl.ac.uk (C.T.); ammar.al-chalabi@kcl.ac.uk (A.A.-C.); 4Perron Institute for Neurological and Translational Science, Ground RR Block QE II Medical Centre Ralph & Patricia Sarich Neuroscience Building, 8 Verdun St, Nedlands, WA 6009, Australia; 5NIHR Maudsley Biomedical Research Centre (BRC), South London and Maudsley NHS Foundation Trust, 16 De Crespigny Park, London SE5 8AF, UK

**Keywords:** Oxford Nanopore Technologies, long-read sequencing, short-read sequencing, variant calling, benchmark, genomic variants, low-complexity regions, multiplexing, experimental variables

## Abstract

Oxford Nanopore Technologies (ONT) long-read sequencing (LRS) has emerged as a promising genomic analysis tool, yet comprehensive benchmarks with established platforms across diverse datasets remain limited. This study aimed to benchmark LRS performance against Illumina short-read sequencing (SRS) and microarrays for variant detection across different genomic contexts and to evaluate the impact of experimental factors. We sequenced 14 human genomes using the three platforms and evaluated single nucleotide variants (SNVs), insertions/deletions (indels), and structural variants (SVs) detection, stratifying by high-complexity, low-complexity, and dark genome regions while assessing effects of multiplexing, depth, and read length. LRS SNV accuracy was slightly lower than that of SRS in high-complexity regions (F-measure: 0.954 vs. 0.967) but showed comparable sensitivity in low-complexity regions. LRS showed robust performance for small (1–5 bp) indels in high-complexity regions (F-measure: 0.869), but SRS agreement decreased significantly in low-complexity regions and for larger indel sizes. Within dark regions, LRS identified more indels than SRS, but showed lower base-level accuracy. LRS identified 2.86 times more SVs than SRS, excelling at detecting large variants (>6 kb), with SV detection improving with sequencing depth. Sequencing depth strongly influenced variant calling performance, whereas multiplexing effects were minimal. Our findings provide valuable insights for optimising LRS applications in genomic research and diagnostics.

## 1. Introduction

Since the completion of the Human Genome Project [1,2], genomics has evolved rapidly, with new technologies enabling the study of various genomic variations. The first human reference genome assembly in 2003 [3] provided the foundation for variant calling methods that facilitated disease-association genomic studies and clinical genetic diagnostics [4]. Three main approaches characterise genomic variants, including DNA microarrays, short-read sequencing (SRS), and long-read sequencing (LRS).

SNP arrays offer a cost-effective tool for detecting single nucleotide variants (SNVs), one of the most common types of variation [5]. The arrays’ high-density oligonucleotide probes accurately identify known SNVs across the genome [6], making them valuable for large-scale population studies and genome-wide association studies (GWAS) [7]. However, microarrays are limited to detecting known variants, restricting their utility for discovering novel or rare variants and structural variations, such as balanced chromosomal rearrangements [6].

SRS has become a cornerstone in genomics research and clinical diagnostics due to its high accuracy (~0.1% error rate) [8], throughput, and cost-effectiveness [9]. Efficient and scalable bioinformatics analysis pipelines have made SRS data analysis increasingly accessible [10,11,12], establishing it as the standard for whole genome studies [13,14] and clinical applications [15,16]. Conversely, SRS’s short read lengths (100–300 bp) pose challenges when analysing repetitive genomic regions, such as segmental duplications [17], transposable elements (TEs) [18], short tandem repeats (STRs) [19], homopolymers [20], and telomeres [21], where reads map ambiguously or fail to map entirely. Despite computational approaches to improve variant calling in these regions [22], many “dark regions” of the genome remain poorly characterised using SRS alone [23].

LRS technologies address many of the SRS limitations in repetitive genomic regions. Oxford Nanopore Technologies (ONT), introduced commercially in 2014, has gained significant attention due to its unique advantages and potential applications. It utilises a fundamentally different approach, passing a single strand of DNA through protein nanopores [24], measuring ionic current disruptions caused by the molecule traversal to determine base sequences [25]. This approach generates exceptionally long reads (typically ranging from 10 to 100 kb, with some exceeding 2 Mb) [26,27], which can span entire repetitive regions, resolve complex structural variants, improve de novo genome assemblies [23], and detect epigenetic marks, such as methylation, without additional library preparation methods [28].

While early ONT LRS iterations had high error rates (38% for R6 nanopore chemistry), based on read-to-reference alignment metrics [29], recent advancements in base-calling algorithms and pore chemistry have substantially improved accuracy, with the latest R10.4 chemistry achieving a modal read accuracy of over 99% (Q20) [30] and an average accuracy of 96.8% (Q15) [31] on read-to-reference alignment. Furthermore, duplex sequencing consensus approaches have further improved accuracy, though with reduced throughput [32]. These advantages position ONT as a valuable tool for comprehensive genome analysis, particularly in low-complexity regions where SRS falls short [33].

Previous LRS benchmarking has primarily used well-characterised Genome in a Bottle (GIAB) reference samples [33,34,35,36,37], but there remains a need for comprehensive comparisons using diverse sample datasets. To address this, we present a new benchmark of LRS using a cohort of 14 in-house sequenced genomes. Our multi-platform approach combines data from Illumina microarrays, Illumina SRS, and ONT LRS, thereby allowing thorough evaluation of ONT’s capabilities in calling SNVs, indels, and structural variants (SVs). By comparing results across these established technologies, we gain insights into each platform’s strengths and limitations across different genomic contexts.

## 2. Results

### 2.1. Sequencing Quality Control

To evaluate the performance of our LRS dataset, we first conducted an analysis of the sequencing data quality and characteristics. We examined several key metrics, including sequencing yields, read length and quality distributions, alignment quality, sequencing depth, and flowcell and barcoding performance, to identify potential factors that may influence variant detection accuracy (Figure 1).

#### 2.1.1. Sequencing Yields

Our analysis of multiplexed (two barcoded samples per flowcell) and singleplexed (one sample per flowcell) nanopore sequencing runs revealed significant differences in yield and read length characteristics between the two approaches.

In terms of sequencing yield, singleplexed samples consistently produced a higher number of reads and bases compared to multiplexed samples (Figure 2A,B). The mean number of reads for singleplexed samples was 9,932,389 (SD: 2,541,662), 118.48% higher than the mean of 4,546,234 (SD: 1,382,487) reads for multiplexed samples. Similarly, the mean number of bases for singleplexed samples (mean: 64,410,968,490; SD: 16,697,589,113) was 99.94% higher than that of multiplexed samples (mean: 32,214,905,852; SD: 5,350,881,127).

The read length distributions showed a different pattern (Figure 2C,D). While the mean read length for multiplexed samples (mean: 7086 bp; SD: 7668) was 8.48% higher than that of singleplexed samples (mean: 6485 bp; SD: 6155), the median read length for singleplexed samples (5369 bp) was 27.08% higher than that for multiplexed samples (4225 bp). The distribution of read lengths across all samples (Figure 2E) reveals that while both approaches produced reads spanning a wide range of lengths, there were notable differences in the distribution patterns. Notably, all singleplexed samples exhibited a clear bimodal distribution of read lengths, characterised by two distinct peaks: one at shorter read lengths (approximately 400 bp) and another at longer read lengths (approximately 10,000 bp). In contrast, the multiplexed samples showed more variable distribution patterns. While some multiplexed samples displayed a bimodal distribution similar to the that of singleplexed samples, although with less pronounced peaks, other multiplexed samples exhibited only a single peak, typically at shorter read lengths (around 1000–5000 bp).

This difference in distribution patterns suggests that singleplexing consistently produced two distinct populations of reads: a shorter fraction and a longer fraction. Multiplexing, on the other hand, appeared to be less consistent in generating the longer read fraction, with some samples failing to produce the second peak at higher read lengths. This might explain why the larger number of reads per flowcell produced in singleplex did not translate into a larger number of genotyped bases.

#### 2.1.2. Read and Alignment Quality

Both read quality and alignment metrics showed minimal differences between multiplexed and singleplexed samples. Base quality distribution (Figure 3A) showed a similar pattern across all samples, with peaks at quality scores 17–22. The mean base quality for singleplexed samples (mean: 20.64; SD: 4.70) was marginally higher than that of multiplexed samples (mean: 20.45; SD: 4.69), representing a small 0.95% increase. Similarly, the median base quality for singleplexed samples (20.60) was slightly higher than that for multiplexed samples (20.34), a 1.24% increase.

The mapping quality distribution (Figure 3B) demonstrated that the vast majority of reads across all samples mapped with a quality score of 60. This pattern was nearly identical between singleplexed and multiplexed samples, with mean mapping qualities of 56.71 (SD: 12.67) and 56.68 (SD: 12.75), respectively, representing a negligible difference of 0.05%. The median mapping quality was 60 for both approaches, indicating no practical difference in mapping performance.

Both multiplexed and singleplexed samples showed a similar trend on the proportion of reads exceeding various quality score thresholds (Q5, Q7, Q10, Q12, and Q15) (Figure 3C,D). While the percentage of reads decreased as the quality threshold increased, at the highest threshold of Q15, all samples retained over 85% of their reads, suggesting multiplexing can be employed without significantly compromising data quality.

#### 2.1.3. Depth of Coverage

Focusing on sequencing depth of coverage per chromosome and across the whole genome, we observed differences between singleplexed and multiplexed samples in line with our expectations. Firstly, per-chromosome sequencing depth (Figure 4A) showed consistent patterns across all chromosomes, with singleplexed samples (mean: 19.01; SD: 5.98) exhibiting higher depth than multiplexed samples (mean: 9.45; SD: 2.33), even though singleplexed samples demonstrated greater variability. The mean whole genome depth (Figure 4B) mirrored the per-chromosome trends. Singleplexed samples achieved a mean depth of 19.84 (SD: 5.22), while multiplexed samples reached 9.84 (SD: 1.58), indicating a 101.63% increase in the mean depth of singleplexed samples. The median depths were 17.20 and 9.93 for singleplexed and multiplexed samples, respectively, showing a 73.30% increase. The consistent depth patterns across chromosomes show that ONT LRS maintains relatively uniform coverage across the genome in both settings, with expected drops in depth for sex chromosomes.

#### 2.1.4. Flowcell and Barcoding Quality

We analysed the initial nanopore availability in the flowcells we used for sequencing to investigate its impact on sequencing performance and depth of coverage obtained. There were distinct differences in the number of pores available at the start of sequencing (Figure 4C), with multiplexed flowcells consistently having a higher number of starting pores (mean: 8116.00; SD: 100.34) compared to singleplexed flowcells (mean: 7008.75; SD: 1190.38), which exhibited greater variability in starting pore numbers. This occurred because the best-quality flowcells were selected for multiplexing to yield the best results possible for this approach. The relationship between initial pore count and total mean whole genome depth (Figure 4D) revealed a strong positive correlation (*r* = 0.64, *p* = 0.034). The linear regression analysis yielded a slope of 0.0025, which indicates that, for every additional 1000 starting pores, we can expect an increase of approximately 2.5× in mean whole genome depth. This suggests that initial pore availability is a significant factor in determining sequencing depth yields.

Examining barcoded read distribution in multiplexed samples allowed us to assess the efficiency of the barcoding process and explain the differences in depth yields between multiplexed and singleplexed flowcells with similar numbers of available nanopores (Figure 4E). Across the three multiplexed flowcells, we observed some variation in the number of reads assigned to the barcodes of interest and unclassified reads. To quantify this variation, we calculated the Coefficient of Variation (CV) for each flowcell: Multiplex Flowcell 1 showed the highest variation (CV: 63.78%), while Multiplex Flowcell 3 exhibited a CV of 38.91%; in contrast, Multiplex Flowcell 2 demonstrated the most consistent read distribution (CV: 31.77%). Importantly, an average of 19.04% (SD: 3.15%) of reads remained unclassified across all multiplexed flowcells. This level of variation within flowcells and the substantial proportion of unclassified reads indicates there is room for efficiency improvement in the barcoding process and, if not optimised, multiplexing results in lower total yield per flowcell.

### 2.2. Single Nucleotide Variants Benchmark

We compared the performance of the ONT and Illumina sequencing platforms for SNV detection with the VCFeval tool of RTG tools v3.12.1 [38,39]. Given their high accuracy in genotyping known SNVs, we used the Illumina microarray datasets as the gold standard, stratified by high-complexity regions (i.e., excluding the Tandem Repeats and Homopolymers regions of GIAB v3.4 [34]) and low-complexity genomic regions (only including the aforementioned sites). In addition, only the sites present in the SNP arrays were retained in SRS and LRS data for comparison. For ONT LRS, we analysed 9,563,097 variants in high-complexity regions and 482,788 variants in low-complexity regions. For SRS, we analysed 9,476,435 variants in high-complexity regions and 473,458 variants in low-complexity regions. Figure 5A,B shows the precision, sensitivity, and F-measure for both technologies.

In high-complexity regions, SRS maintains an advantage over LRS technology, with statistically significant differences (*p* < 0.01) in mean precision (SRS: 0.963, SD = 0.004; LRS: 0.953, SD = 0.010), sensitivity (SRS: 0.972, SD = 0.001; LRS: 0.955, SD = 0.021), and F-measure (SRS: 0.967, SD = 0.002; LRS: 0.954, SD = 0.015) compared to LRS. However, the performance slightly shifted in low-complexity regions. While SRS still exhibited higher mean precision (SRS: 0.799, SD = 0.005; LRS: 0.778, SD = 0.013, *p* < 0.001) and F-measure (SRS: 0.779, SD = 0.003; LRS: 0.764, SD = 0.016, *p* < 0.004), the gap in performance narrowed considerably for mean sensitivity (SRS: 0.760, SD = 0.002; LRS: 0.751, SD = 0.019), with no statistically significant difference (*p* > 0.05) between platforms. Overall, the performance of both technologies degrades in low-complexity regions, but to different degrees, with a more pronounced performance drop for SRS.

To further elucidate the performance of each technology, we conducted an in-depth analysis of SNV error rates, stratified by complexity and error type (Figure 5C–F), revealing distinct error profiles for SRS and LRS across different SNV types. In high-complexity regions, LRS exhibited significantly higher error rates across all SNV types for both false positives (FP) and false negatives (FN) (*p* < 0.05 for all comparisons, Figure 5C,D). These significant differences were observed across both transitions and transversions. The error profile shifted in low-complexity regions (Figure 5E,F). While for FPs, LRS maintained significantly higher error rates across all SNV types (*p* < 0.001), for FNs in low-complexity regions, LRS demonstrated significantly higher error rates specifically for the transversions A > C, C > A, G > C, and G > T (*p* < 0.05), though the remaining SNV types showed no significant differences between technologies.

Therefore, while SRS generally maintains an advantage in high-complexity regions and in minimising false positives, LRS shows competitive performance in detecting certain SNV types within low-complexity regions, particularly in terms of false negative rates. The observed differences in error profiles between the two technologies likely stem from their fundamental sequencing approaches. SRS’s short-read methodology may struggle with repetitive regions due to mapping ambiguities, while LRS’s long-read approach can span these regions but introduces more base-calling errors.

### 2.3. Indels Benchmark

To evaluate the performance of LRS for indel detection using VCFeval, we utilised the SRS dataset as the gold standard, as SRS has demonstrated superior performance in indel detection with F1 scores of up to 0.99 for state-of-the-art variant calling pipelines [40]. However, in the following analyses, it is important to note that SRS may become a suboptimal gold standard in low-complexity regions and as indel length increases.

Our results showed a marked disparity in performance between high- and low-complexity regions (Figure 6A,B). On one hand, in high-complexity regions, LRS exhibited robust performance with a mean precision of 0.786 (SD = 0.067), sensitivity of 0.904 (SD = 0.035), and F-measure of 0.841 (SD = 0.053). Conversely, in low-complexity regions, we observed a significant decline in performance metrics, with mean precision, sensitivity, and F-measure dropping to 0.411 (SD = 0.063), 0.495 (SD = 0.045), and 0.449 (SD = 0.056), respectively.

We further analysed the size distribution of insertions and deletions detected by both LRS and SRS platforms in high- and low-complexity genomic regions (Figure 6C–F), to ascertain potential biases in the detection of indels of various sizes. We classified the variants into insertions and deletions, within their respective complexity regions. Indeed, both technologies performed comparably in detecting small indels within high-complexity genomic regions (Figure 6C,E), with similar size distributions peaking at 1–2 bp and rapidly decreasing for larger sizes (KS statistic: 0.013 for insertions, 0.013 for deletions). In low-complexity regions (Figure 6D,F), LRS demonstrated a higher propensity for detecting larger indels, particularly around 8–12 bp, although with small effect sizes (KS statistic: 0.028 for insertions, 0.012 for deletions). All differences were statistically significant (*p* << 0.001 after FDR correction).

To investigate the performance characteristics of LRS across different indel lengths, we stratified the analysis by indel size categories and complexity regions (Figure 6G–I). In high-complexity regions, LRS demonstrated robust performance for smaller indels (1–5 bp), which represented most variants (89.5%) in these regions, achieving high precision (median = 0.826, IQR = 0.071), sensitivity (median = 0.916, IQR = 0.040), and F-measure (median = 0.868, IQR = 0.066). Performance remained stable for medium-sized indels (6–20 bp), with only modest decreases in precision (median = 0.772–0.798) and sensitivity (median = 0.898–0.932). However, there was a notable decline in performance for larger indels (21–50 bp), particularly in precision (median = 0.624, IQR = 0.184), while maintaining relatively high sensitivity (median = 0.884, IQR = 0.100).

In low-complexity regions, we observed substantially lower performance across all size categories. For the predominant 1–5 bp small indels (83.5% of variants), precision (median = 0.445, IQR = 0.116) and sensitivity (median = 0.483, IQR = 0.109) were markedly reduced compared to high-complexity regions. Performance further deteriorated with increasing indel size, with the largest indels (21–50 bp) showing the lowest precision (median = 0.250, IQR = 0.093), maintaining moderate sensitivity (median = 0.572, IQR = 0.085).

Stretched exponential regression analysis revealed strong relationships between indel size and performance metrics in high-complexity regions (*R*^2^ = 0.849, 0.672, and 0.917 for precision, sensitivity, and F-measure, respectively), while relationships were weaker or absent in low-complexity regions (*R*^2^ = 0.690, 0.000, and 0.575, respectively). Therefore, this confirms that indel size significantly impacts variant calling accuracy, with the effect being more pronounced in high-complexity regions.

This size-dependent performance pattern highlights the ongoing challenges in accurately sequencing and analysing repetitive or structurally complex genomic areas. Moreover, when considering the inherent limitations of our benchmarking approach, it is important to interpret these results with caution. While we used Illumina SRS as the gold standard for this comparison, it is important to note that SRS itself has limitations, particularly in sequencing repetitive regions. The increasing error rates with increasing indel size could reflect the limitations of SRS in detecting larger indels, especially in repetitive regions. Indeed, some of the false positives in LRS, particularly for larger indels, may actually represent true variants that were missed by SRS.

### 2.4. Impact of Multiplexing on Variant Calling

We compared the performance of variant calling, for both SNVs and indels in high-complexity and low-complexity regions, for multiplexed and singleplexed samples, to determine the effect of multiplexing on the accuracy of variant calling.

For SNVs in high-complexity regions (Figure 7A), singleplexed samples demonstrated marginally superior performance across all metrics. Indeed, mean precision (singleplex: 0.960, SD = 0.003; multiplex: 0.944, SD = 0.008), sensitivity (singleplex: 0.969, SD = 0.004; multiplex: 0.935, SD = 0.018), and F-measure (singleplex: 0.965, SD = 0.004; multiplex: 0.939, SD = 0.013) were all higher for singleplexed samples. A similar trend was evident in low-complexity regions (Figure 7B), with higher mean precision (singleplex: 0.788, SD = 0.005; multiplex: 0.765, SD = 0.009), sensitivity (singleplex: 0.764, SD = 0.005; multiplex: 0.733, SD = 0.016), and F-measure (singleplex: 0.776, SD = 0.005; multiplex: 0.749, SD = 0.012) for singleplexed samples.

Indel calling exhibited more pronounced differences between multiplexing conditions. In high-complexity regions (Figure 7C), singleplexed samples showed substantial improvements in mean precision (singleplex: 0.836, SD = 0.028; multiplex: 0.720, SD = 0.035), sensitivity (singleplex: 0.929, SD = 0.008; multiplex: 0.872, SD = 0.030), and F-measure (singleplex: 0.880, SD = 0.019; multiplex: 0.789, SD = 0.033). The disparity was even more evident in low-complexity regions (Figure 7D), with higher mean precision (singleplex: 0.456, SD = 0.044; multiplex: 0.352, SD = 0.018), sensitivity (singleplex: 0.528, SD = 0.022; multiplex: 0.452, SD = 0.025), and F-measure (singleplex: 0.489, SD = 0.034; multiplex: 0.395, SD = 0.021) for singleplexed samples.

To separate the effects of multiplexing and sequencing depth, we performed an analysis of covariance (ANCOVA) (Figure 8). This analysis revealed that sequencing depth had a significant positive effect on all performance metrics (*p* < 0.05) in high- and low-complexity regions. The impact of depth was particularly pronounced for indel calling, as evidenced by both highly significant *p*-values (*p* < 0.01) and substantially larger effect size estimates compared to those observed for SNV calling. The independent multiplexing effect, after controlling for depth, was not statistically significant (*p* > 0.05) across any metric, suggesting that the apparent performance differences observed in the initial comparisons were primarily driven by variations in sequencing depth rather than an independent effect of multiplexing itself.

Thus, while initial comparisons suggested that multiplexing has a modest impact on SNV calling performance and more significantly affects indel calling accuracy, our subsequent analysis indicates that these effects are largely attributable to differences in sequencing depth rather than multiplexing per se. These findings suggest that maintaining adequate sequencing depth is more critical for variant calling performance than the choice between multiplexed and singleplexed approaches, though the impact varies considerably across genomic contexts.

### 2.5. Impact of Sequencing Depth on Variant Calling

To assess the influence of sequencing depth on variant calling performance, we analysed the relationship between whole genome mean depth and the variant calling performance metrics (precision, sensitivity, and F-measure) for both SNVs and indels in high-complexity and low-complexity regions.

For SNVs in high-complexity regions (Figure 9A), we observed strong positive correlations between sequencing depth and all performance metrics (precision: *r* = 0.81, *p* = 4.94 × 10^−4^; sensitivity: *r* = 0.81, *p* = 3.89 × 10^−4^; F-measure: *r* = 0.82, *p* = 3.26 × 10^−4^). Similar trends were observed for SNVs in low-complexity regions (Figure 9B), with slightly stronger correlations (precision: *r* = 0.86, *p* = 6.68 × 10^−5^; sensitivity: *r* = 0.83, *p* = 2.73 × 10^−4^; F-measure: *r* = 0.85, *p* = 9.99 × 10^−5^).

Indel calling performance showed even stronger correlations with sequencing depth. In high-complexity regions (Figure 9C), the correlations were highly significant for all metrics (precision: *r* = 0.93, *p* = 1.45 × 10^−6^; sensitivity: *r* = 0.83, *p* = 2.03 × 10^−4^; F-measure: *r* = 0.91, *p* = 7.59 × 10^−6^), but the strongest correlations were observed for indels in low-complexity regions (Figure 9D) (precision: *r* = 0.99, *p* = 5.44 × 10^−12^; sensitivity: *r* = 0.95, *p* = 3.53 × 10^−7^; F-measure: *r* = 0.98, *p* = 6.27 × 10^−10^).

Overall, the relationship between sequencing depth and performance metrics followed an asymptotic trend for all variant types and genomic regions, therefore suggesting that while increasing sequencing depth generally improves variant calling performance, the rate of improvement plateaus at higher depths. Additionally, the impact of sequencing depth is more pronounced for indel calling compared to SNV calling, particularly in low-complexity regions.

### 2.6. Impact of Read Length on Variant Calling

To investigate the relationship between read length and variant calling performance, we analysed the correlation between mean read length and the three performance metrics: precision, sensitivity, and F-measure, for both SNVs and indels, in high-complexity and low-complexity regions.

For SNVs in high-complexity regions (Figure 10A), we observed moderate negative correlations between mean read length and all three performance metrics (precision: *r* = −0.43, *p* = 1.26 × 10^−1^; sensitivity: *r* = −0.51, *p* = 6.00 × 10^−2^; F-measure: *r* = −0.49, *p* = 7.38 × 10^−2^). Similar negative correlations were observed for SNVs in low-complexity regions (Figure 10B) (precision: *r* = −0.41, *p* = 1.48 × 10^−1^; sensitivity: *r* = −0.50, *p* = 6.64 × 10^−2^; F-measure: *r* = −0.47, *p* = 8.76 × 10^−2^).

Moreover, indel calling performance also showed comparable negative correlations with mean read length. In high-complexity regions (Figure 10C), the correlations were slightly stronger than those observed for SNVs (precision: *r* = −0.48, *p* = 8.11 × 10^−2^; sensitivity: *r* = −0.51, *p* = 6.34 × 10^−2^; F-measure: *r* = −0.49, *p* = 7.19 × 10^−2^), and for indels in low-complexity regions (Figure 10D), the correlations were also negative (precision: *r* = −0.40, *p* = 1.61 × 10^−1^; sensitivity: *r* = −0.46, *p* = 9.48 × 10^−2^; F-measure: *r* = −0.42, *p* = 1.31 × 10^−1^).

None of these observed correlations for SNVs or indels in high- or low-complexity regions reached statistical significance (*p* > 0.05). Therefore, these results suggest a trend toward longer read lengths being associated with a moderate decrease in variant calling performance for both SNVs and indels, although this trend was not statistically significant in this dataset. This stands in contrast to the significant impact of sequencing depth on variant calling described in the previous section.

### 2.7. Dark Genome Analysis

To specifically evaluate variant calling performance within challenging genomic regions, we analysed the mean number of SNVs and indels detected by LRS and SRS technologies within the “dark genome” regions defined by Ebbert et al. [23]. In this analysis, we excluded microarrays due to the difficulty genotyping these regions with this technology [41], which renders it unsuitable as a gold standard. We applied standard quality filters (Q20 and Q30) to assess the proportion of high-quality variants retained by each platform (Figure 11).

For SNV detection (Figure 11A), SRS detected a higher number of SNVs (170.21; SD: 13.99), compared to LRS, with 168.50 (SD: 13.33), or 99.02%, passing a QUAL score > 20 and 167.71 (SD: 12.93), or 98.57%, passing QUAL > 30. Long-read sequencing identified a lower mean number of SNVs (94.64; SD: 14.73). Quality filtering had a substantial impact on LRS calls, retaining only 40.14 (SD: 9.05), or 42.39%, of variants at QUAL > 20 and 15.50 (SD: 6.61), or 16.26%, at QUAL > 30. In the analysis of indels (Figure 11B), LRS generated a markedly higher mean number of calls (26,942.79; SD: 1038.38) compared to SRS (3523.07; SD: 141.26). However, similarly to SNVs, a large proportion of LRS indel calls had lower quality scores, as only 9385.29 (SD: 1676.28), or 34.68%, of LRS indels passed QUAL > 20, and 3700.00 (SD: 1273.55), or 13.60%, passed QUAL > 30. In contrast, the vast majority of SRS indel calls were high quality, with 3494.21 (SD: 138.37), or 99.18%, passing QUAL > 20 and 3487.79 (SD: 138.17), or 99.00%, passing QUAL > 30.

These findings suggest that while LRS has the potential to identify a larger number of indels in dark regions compared to SRS, a substantial fraction of these initial calls are of lower quality based on standard metrics. The number of high-quality (QUAL > 30) indels detected by LRS is only marginally higher than the total number of high-quality indels detected by SRS, highlighting the trade-off between quantity and quality in these challenging genomic contexts and aligning with the lower precision observed for LRS indels in Section 2.3.

### 2.8. Structural Variant Calling Evaluation

To evaluate the performance of ONT LRS and Illumina SRS platforms in detecting structural variants (SVs), we conducted a consensus analysis with SURVIVOR [42] to merge consensus SV calls from both technologies. SV calls were considered in consensus if they met the following criteria: (1) located within 500 base pairs of each other, (2) detected in both VCF files for a given sample, (3) classified as the same SV type, (4) identified on the same strand, and (5) exhibited a size difference of no more than 30% between platforms.

Overall, ONT LRS identified 2.86 times more SVs compared to Illumina SRS across all samples (Figure 12A). However, the consensus between the two platforms was relatively low, with an average of 20.61% (SD = 1.60%) of LRS calls and 57.87% (SD = 10.11%) of SRS calls being concordant. Furthermore, regarding SV size distribution (Figure 12B), LRS demonstrated a broader range of SV sizes, particularly in detecting larger variants, while Illumina SRS showed a bias towards detecting smaller SVs. In fact, LRS detected SVs ranging from 12 bp to 129,371,498 bp, with a mean size of 3527.59 bp (SD = 482,253.95 bp) and a median of 80 bp. In contrast, SRS identified SVs between 2 bp and 6064 bp, with a mean size of 165.62 bp (SD = 164.47 bp) and a median of 96 bp. The markedly larger maximum SV size (129 Mb for LRS vs. 6 Kb for SRS) and higher SD observed in LRS data highlight its superior capability in detecting large-scale genomic rearrangements, which are often challenging to identify with SRS.

To further investigate the nature of these SV calls, we analysed the distribution of SV types for each sequencing platform (Figure 12C,D). For LRS, insertions were the most frequently detected (mean = 14,784.64, SD = 3352.94, per sample), followed closely by deletions (mean = 12,074.57, SD = 2675.39). SRS, however, detected deletions most frequently (mean = 5491.57, SD = 139.84), followed by insertions (mean = 3230.07, SD = 127.40). Notably, SRS detected a higher number of breakends (BND) (mean = 741.21, SD = 44.07) compared to LRS (mean = 53.14, SD = 35.98), while LRS showed the capability to detect STRs (mean = 25.64, SD = 8.21).

The size distribution of different SV types further showed distinct patterns between LRS and SRS platforms. For insertions (Figure 13A), both technologies showed multimodal distributions, with LRS capturing a broader range of sizes. SRS exhibited a trimodal distribution with a sharp peak at smaller insertion sizes (<10 bp), a second peak around 100 bp, and a third, broader peak centred approximately at 300 bp. In contrast, LRS demonstrated a bimodal distribution with peaks at approximately 300 bp and 6 kb. Deletion size distributions (Figure 13B) showed similar bimodal patterns for both platforms, with LRS again exhibiting a wider range. SRS deletions were predominantly clustered below 1 kb, while LRS effectively captured deletions spanning several orders of magnitude, from 50 bp to 10 Mb.

Duplication events (Figure 13C) were primarily detected by LRS, with sizes ranging from 100 bp to 100 Mb and a peak around 10 kb. The absence of a substantial SRS signal for duplications highlights the advantage of LRS in identifying these complex structural variants. Inversion size distributions (Figure 13D) showed striking differences between the platforms. LRS detected inversions across a broad size range (10^2^ to 10^8^ bp), while SRS mainly identified smaller inversions (<10^3^ bp).

Breakend (BND) size distributions (Figure 13E) were similar between platforms, with both showing a peak around 20 bp. However, SRS demonstrated a slightly broader range, extending to approximately 100 bp. Short tandem repeat (STR) size distributions (Figure 13F) were exclusively captured by LRS, with sizes ranging up to approximately 700 bp and a peak around 50 bp.

To investigate the genomic distribution of SVs, we analysed their occurrence across chromosomes, normalizing for chromosome length (Figure 14A). For both LRS and SRS platforms, chromosomes 19, 20, and 21 exhibited notably higher percentages of SVs per megabase compared to other chromosomes, while, conversely, chromosomes 14, 15, X, and Y showed lower SV densities for both platforms. Interestingly, despite their technological differences, both LRS and SRS platforms showed similar trends in chromosomal SV distribution. A further correlation analysis between chromosome length and SV occurrence (Figure 14B) revealed weak and non-significant correlations between chromosome length and SV counts for both LRS (*r* = 0.11, *p* = 0.616) and SRS (*r* = 0.05, *p* = 0.824) platforms, suggesting that chromosome length is not a strong predictor of SV abundance.

The elevated SV density observed in chromosomes 19, 20, and 21 could be attributed to various factors, including higher gene density, increased recombination rates, or the presence of specific genomic features that promote structural rearrangements [43]. For instance, chromosome 19 is known for its high gene density and elevated GC content, which may contribute to its increased propensity for structural variations [44].

### 2.9. Impact of Sequencing Depth on Calling Structural Variants

To evaluate the influence of sequencing depth on structural variant (SV) detection, we analysed the relationship between whole genome mean depth and both the number and size distribution of SV calls across LRS and SRS platforms.

For LRS, we observed a strong positive correlation between sequencing depth and the number of SV calls (Figure 15A) (*r* = 0.94, *p* = 6.71 × 10^−7^). The relationship followed an asymptotic trend, with the number of SV calls increasing from approximately 16,000 at 10× coverage to around 35,000 at 30× coverage, though the rate of increase diminished at higher depths. The consensus calls between the two platforms demonstrated a strong positive correlation with sequencing depth (Figure 15B) (*r* = 0.89, *p* = 2.14 × 10^−5^), suggesting that increased depth primarily improves the detection of true structural variants. Analysis of SV size distributions across sequencing depths revealed that for LRS, we observed consistent detection of SVs across a broad size range (10^1^ to 10^8^ bp) at all sequencing depths, with no significant trend in median SV size (*r* = 0.006, *p* = 2.17 × 10^−4^). The mean SV size showed considerable variation (range: 412–9848 bp), likely due to improved detection of larger variants at higher depths (Figure 15C).

These findings suggest that while increased sequencing depth substantially improves SV detection in long-read sequencing, particularly for larger variants, short-read SV calling remains relatively consistent across different depths. The data indicate that 25× coverage may be optimal for long-read SV detection, as the rate of novel SV discovery plateaus beyond this depth.

## 3. Discussion

This study provides a comprehensive comparison of Oxford Nanopore Technologies (ONT) long-read sequencing (LRS) and Illumina short-read sequencing (SRS) platforms, focusing on their performance in variant detection across high- and low-complexity regions, as well as a dedicated analysis within the challenging “dark genome” regions defined by Ebbert et al. [23]. Using an in-house cohort of 14 genomes, rather than standard reference samples, this multi-platform approach on diverse samples allows a direct, comparative assessment of SNV, indel, and SV detection capabilities across varying genomic contexts, providing new insights into the practical performance of LRS and the impact of experimental factors, such as multiplexing and sequencing depth.

### 3.1. Sequencing Quality and Yields

The comparison between multiplexed and singleplexed LRS samples revealed significant differences in sequencing yield and read length, with singleplexed samples producing approximately twice the number of reads and bases. While multiplexing offers cost savings and increased throughput, our results suggest that it may come at the expense of reduced per-flowcell sequencing depth. This reduction in depth can influence variant detection accuracy, although our ANCOVA analysis indicated depth itself was the primary driver of performance differences observed between the two approaches.

Interestingly, we observed distinct read length distribution patterns between multiplexed and singleplexed samples. The consistent bimodal distribution in singleplexed samples, with peaks at shorter (~400 bp) and longer (~10,000 bp) read lengths, contrasts with the more variable patterns seen in multiplexed samples. This difference may reflect variations in library preparation or sequencing efficiency between the two approaches and warrants further investigations to optimise multiplexing protocols for long-read sequencing.

### 3.2. Single Nucleotide Variants Detection

For SNVs, our findings revealed that SRS maintains a slight edge in high-complexity regions, achieving higher precision, sensitivity, and F-measure (mean F-measure 0.967 vs. 0.954 for LRS). However, LRS performance becomes more competitive in low-complexity regions, particularly regarding sensitivity, where the difference between LRS (0.751) and SRS (0.760) was not statistically significant. Both technologies showed performance degradation in low-complexity regions, but the drop was more pronounced for SRS, particularly in sensitivity. The error profile analysis highlighted that LRS generally exhibited higher error rates (both FP and FN) in high-complexity regions. In low-complexity regions, while LRS still had higher FP rates, the FN rates became more comparable for several transversion types, suggesting LRS’s long reads may help overcome some mapping ambiguities that challenge SRS, even if base-calling accuracy remains a factor.

### 3.3. Indel Detection

The performance of LRS, with respect to SRS, in indel detection showed marked differences between high- and low-complexity genomic contexts. In high-complexity areas, LRS demonstrated robust F-measures of up to 0.869 for small indels (1–5 bp), which constitute the majority of these variants. Performance remained relatively stable for medium-sized indels (6–20 bp) but declined for larger ones (21–50 bp), especially in precision. However, performance dropped significantly in low-complexity regions across all indel sizes, with the median F-measure for small indels falling to 0.445.

This size-dependent pattern highlights the ongoing challenges in accurately calling indels, especially in repetitive areas. While the particularly poor performance for larger indels in low-complexity regions could partially reflect limitations in our SRS-based gold standard, as SRS itself struggles with larger indels in such regions, the overall trend suggests room for improvement in LRS indel calling algorithms. This contrasts with the mapping advantage of LRS in these regions but reflects current challenges in precise variant calling post-alignment. Future benchmarking incorporating orthogonal validation or assembly-based comparisons may help disentangle technological limitations from benchmark artifacts.

### 3.4. Dark Genome Variant Calling Performance

Our specific analysis of variant calling within dark genome regions, originally defined due to challenges for SRS mapping and variant calling [23,45], revealed significant challenges for high base-level accuracy, particularly for LRS, despite its known mapping advantages in such areas. For SNVs, LRS identified the fewest variants initially, and stringent quality filtering retained only ~16% of calls at QUAL > 30, compared to ~99% retention for SRS calls. This resulted in a very low yield of high-quality LRS SNVs at these specific loci. For indels, LRS generated a vastly higher number of calls than SRS in dark regions, but again, stringent filtering (only ~14% passed QUAL > 30) brought the final count of high-quality LRS indels into a range comparable to the high-quality indel count from SRS. This finding, where LRS required stringent filtering to yield high-confidence SNV and indel calls, aligns with the known difficulties in achieving high base-level accuracy within highly repetitive sequences [46,47].

While LRS has the potential to resolve the mapping issues in these regions, our results underscore that translating this improved mappability into high-confidence base-level variant calls (SNVs/indels) within the dark genome remains challenging when assessing it against SRS. However, in interpreting these results, it is important to consider that we rely solely on variant quality scores to filter for high-confidence variants. Such metrics are, in turn, largely dependent on mapping quality scores, which are less reliable in repetitive and dark regions of the genome for SRS [23]. Such a factor could play a role in our experiment and result in an overestimation of SRS high–confidence calls. Nevertheless, LRS’s fundamental advantage in spanning these regions remains critical for structural variation analysis even if achieving high base-level accuracy at specific SNV/indel loci within them currently lags behind SRS calls or requires very stringent filtering.

### 3.5. Impact of Experimental Factors on Variant Calling

Our investigation into the effects of multiplexing, sequencing depth, and read length yielded several important practical insights. While initial comparisons suggested multiplexing negatively impacted variant calling performance, our ANCOVA revealed these effects were largely driven by the resulting lower sequencing depth in multiplexed runs. After controlling for depth, the independent effect of multiplexing was not significant. This suggests researchers can confidently employ multiplexing without inherently compromising variant calling accuracy, provided adequate sequencing depth is maintained. Sequencing depth itself showed a strong, positive, but asymptotic relationship with performance for both SNVs and indels across complexity regions, highlighting diminishing returns at very high coverage and aiding cost–benefit optimisation. The impact of depth was particularly pronounced for indels in low-complexity regions. In contrast, mean read length showed only weak, non-significant negative correlations with variant calling performance in our dataset, suggesting that achieving sufficient depth is currently more critical than maximising read length for SNV/indel accuracy.

### 3.6. Structural Variants Detection

Contrasting with the SNV and small indel benchmarks, our SV analysis demonstrates the strength of LRS in accessing and characterising complex genomic features. LRS identified 2.86 times more SVs than SRS and captured a vastly broader spectrum of structural variations, detecting events up to 129 Mb compared to SRS’s ~6 kb limit. This underscores LRS’s unique ability to uncover large-scale rearrangements often missed by SRS, a capability directly linked to its long reads spanning repetitive and complex regions. This advantage remained consistent across LRS sequencing depths, although the total number of SV calls increased strongly with depth (*r* = 0.94), appearing to plateau around 25–30× coverage, thus suggesting an optimal coverage target.

The distinct patterns observed in SV size distributions and types between LRS and SRS platforms further emphasise their complementary nature. LRS excelled at detecting large insertions, deletions, inversions, and duplications, and exclusively identified STR expansions, repetitive variants often inaccessible to SRS. This superior performance in SV detection aligns with the established strengths of LRS and directly addresses the limitations of SRS in dark and complex genomic regions. It demonstrates that even if base-level accuracy benchmarks (like our SNV/indel analysis) show limitations in specific contexts (like microarray sites in dark regions), the ability of LRS to span repeats and map uniquely provides invaluable information about structural genomic architecture. It is important to note that the choice of SV caller also influences results, and future comparative studies should incorporate multiple callers for both LRS and SRS.

In addition, the analysis of chromosomal SV distribution revealed hotspots on chromosomes 19, 20, and 21 for both platforms, correlating with known features like high gene density [43,44], indicating biological drivers alongside technological detection capabilities.

### 3.7. Limitations and Future Directions

While our study provides valuable insights into the performance of ONT LRS, several limitations should be acknowledged. The non-random allocation of higher-quality flowcells to multiplexing experiments introduced a methodological bias, potentially masking the true impact of multiplexing beyond depth. Moreover, our sample size (*n* = 14) limits generalisability to broader population variations, although it proved sufficient to consistently observe platform-specific performance trends and the impact of experimental factors, such as sequencing depth, within this cohort. Crucially, the use of microarrays (for SNVs) and SRS (for indels) as gold standards introduces inherent limitations. These benchmarks may underestimate LRS performance where the gold standard itself fails, particularly for larger variants or within highly repetitive sequences characteristic of low-complexity regions. Additionally, the choice of specific SV callers also impacts detection profiles.

Future studies should employ randomised flowcell allocation for multiplexing experiments and larger, diverse cohorts. Crucially, evaluating LRS performance in dark regions requires benchmarks beyond standard microarrays or SRS comparisons for SNVs/indels. Assembly-based validation or orthogonal technologies might provide a truer picture of accuracy in these challenging areas, complementing the clear advantages LRS demonstrates in mapping and SV detection within them. Comparing multiple variant callers for both LRS and SRS would also yield a more complete picture of SV detection capabilities.

Looking ahead, integrating LRS and SRS presents exciting prospects for comprehensive genomic analysis. Their complementary strengths can provide a more complete picture of genomic variations. Continued improvements in LRS basecalling and variant detection algorithms will be vital in fully realising its potential, particularly for applications such as clinical diagnostics and population screening. In these contexts, achieving very high sensitivity and specificity is crucial, as even seemingly small performance deficits can drastically reduce the positive predictive value for rare diseases, thus complicating clinical interpretation [48]. Furthermore, optimising multiplexing protocols and sequencing strategies will enhance LRS’s cost-effectiveness and accessibility for large-scale genomic studies.

## 4. Materials and Methods

### 4.1. Sample Selection and Preparation

Samples used in this study were obtained from Project MinE, an international collaboration focused on whole-genome sequencing of amyotrophic lateral sclerosis (ALS) patients and controls [49]. From this cohort, we selected 14 samples for analysis. Venous blood was drawn from participants, and genomic DNA was isolated using standard methods. DNA concentrations were standardized to 100 ng/μL as measured by fluorometry using the Quant-iT™ PicoGreen™ dsDNA quantitation assay (Thermo Fisher Scientific, Waltham, MA, USA). DNA integrity was assessed using gel electrophoresis.

### 4.2. Microarray Genotyping

DNA samples were genotyped using the Infinium Omni2.5-8 v1.4 array (Illumina, Cambridge, UK) according to the manufacturer’s protocol. To ensure consistency across platforms, the genotyped variants were lifted over to the GRCh38 reference genome using GATK LiftoverVcf (Picard) v3.1.1 [50]. Only variants that were successfully lifted over and had matching alleles in the reference genome were retained for subsequent analyses.

### 4.3. Illumina Short-Read Sequencing

DNA samples were sequenced using Illumina’s FastTrack services (Illumina, San Diego, CA, USA) on the Illumina HiSeq 2000 platform. Sequencing was performed using 100 bp paired-end reads with PCR-free library preparation, yielding approximately 35× coverage across each sample. The Isaac pipeline [51] was used for alignment to the hg19 reference genome and to call single nucleotide variants (SNVs), insertions and deletions (indels), and larger structural variants (SVs). To ensure consistency across platforms, the called variants were lifted over to the GRCh38 reference genome using GATK LiftoverVcf [50]. Only variants that were successfully lifted over and had matching alleles in the reference genome were retained for subsequent analyses.

### 4.4. Oxford Nanopore Long-Read Sequencing

For ONT sequencing, DNA libraries were extracted with the Monarch^®^ Spin gDNA Extraction Kit T3010S (New England Biolabs, Hitchin, UK) and prepared using the SQK-LSK114 Ligation Sequencing Kit (Oxford Nanopore Technologies, Oxford, UK), according to the respective manufacturers’ protocols. The prepared libraries were quantified using a Qubit fluorometer (Thermo Fisher Scientific). For multiplexed runs, barcoding was performed using the SQK-NBD114.24 Native Barcoding Kit (Oxford Nanopore Technologies). Sequencing was performed on PromethION 24 instruments using R10.4.1 flowcells. Two samples were pooled and sequenced on a single flowcell for multiplexed runs, while individual samples were sequenced on dedicated flowcells for singleplexed runs. Sequencing runs were performed for approximately 72 h.

Raw signal data were base-called with the EPI2ME wf-basecalling v1.4.5 pipeline [52], using the Dorado model dna r10.4.1 e8.2 400 bps sup v5.0.0. The unaligned BAM output was assessed for quality with NanoPlot v1.42.0 [53]. Using the EPI2ME wf-human-variation v2.6.0 pipeline [54,55,56,57,58,59,60,61,62], reads were aligned to the GRCh38 reference genome; this pipeline uses Clair3 [57] to call SNVs and indels, Sniffles2 [58] to detect SVs, and Straglr [62] for STRs.

### 4.5. Variant Comparison and Benchmarking

All variant files were filtered and compared using a custom Nextflow pipeline (Appendix A), and can be executed using the command nextflow run KHP-Informatics/ont-benchmark.

#### 4.5.1. Single Nucleotide Variants and Indels

For SNVs, variants from all three platforms (ONT LRS, Illumina SRS, and Illumina microarray) were intersected with regions excluding and including only the Tandem Repeats and Homopolymers regions defined by the GIAB v3.4 benchmark [34], to form the high-complexity and low-complexity regions, respectively. Only variants present in the microarray dataset were retained for the LRS and SRS datasets to ensure consistent comparison across platforms.

For indels, variants from LRS and SRS platforms were also filtered, based on the same high-confidence regions, into high-complexity and low-complexity sets based on their genomic location.

Comparison of SNVs and indels was performed using RTG Tools vcfeval v3.12 [38,39], to generate true positive, false positive, and false negative variant sets, as well as performance metrics including precision, recall, and F-measure. For SNV comparisons, the microarray data served as the truth set, while for indel comparisons, the Illumina SRS data were used as the truth set.

#### 4.5.2. Dark Genome Variants

SNVs and indels from LRS and SRS were subsetted to include only variants located within the dark genome coordinates defined by Ebbert et al. [23]. For each sample and technology, the total number of SNVs and indels passing standard quality filters (i.e., marked as ‘PASS’ in the VCF FILTER field) within the dark regions was determined. Subsequently, stricter quality thresholds based on the VCF QUAL score were applied: QUAL > 20 and QUAL > 30. The number of variants meeting each threshold was counted separately for SNVs and indels for both LRS and SRS platforms.

#### 4.5.3. Structural Variants

SVs from LRS and SRS were filtered for quality, whereby LRS variants with at least 5 supporting reads and STRs with a minimum of 10 spanning reads were retained, and SRS variants with at least 5 supporting paired-end or split reads were retained.

The filtered callsets were compared using SURVIVOR v1.0.7 [42], by generating a merged consensus set of variants from both platforms which included only variants within 500 base pairs of each other, of the same type, on the same strand, and with a size difference of no more than 30%.

### 4.6. Statistical Analysis

Statistical analyses were performed using jupyterlab 4.3.5 [63], Python 3.12.8 [64], numpy 2.2.2 [65], polars 1.21.0, matplotlib 3.10.0 [66], seaborn 0.13.2 [67], SciPy 1.15.1 [68], scikit-learn 1.6.1 [69], and statsmodels 0.14.4 [70]. This work was performed on the King’s Computational Research, Engineering and Technology Environment (CREATE) at King’s College London [71]. Statistical significance was deemed as a *p*-value < 0.05.

## 5. Conclusions

This study provides a comprehensive benchmark of ONT LRS against established Illumina SRS and microarray platforms using an in-house cohort of 14 human genomes. Our findings highlight the distinct strengths and weaknesses of each technology across different variant types and genomic contexts.

While SRS maintains a slight advantage in SNV accuracy within high-complexity genomic regions, LRS demonstrates comparable sensitivity in low-complexity regions. For indel detection, LRS performed robustly for small variants (1–5 bp) in high-complexity regions, but its accuracy diminished significantly in low-complexity regions and for larger indels, although interpretation requires caution due to the limitations of the SRS gold standard in these contexts. In the dark genome, while LRS can map reads, achieving high-confidence SNV and indel calls benchmarked against microarray/SRS sites requires stringent filtering, leading to low yields compared to raw calls, highlighting current challenges in base-level accuracy within these complex sequences using standard benchmarks.

In contrast, the most significant advantage of LRS was observed in structural variant (SV) detection, where it identified nearly three times more SVs than SRS, excelled at characterising large variants (>6 kb), and uniquely detected STR expansions. This capability highlights the power of LRS in resolving complex genomic architectures, particularly within regions inaccessible to SRS.

Our analysis of experimental factors revealed that sequencing depth is a critical determinant of variant calling performance, especially for indels and SVs, with diminishing returns suggesting optimal coverage levels around 25–30× for SVs. Importantly, the effects of multiplexing were found to be minimal after accounting for sequencing depth, supporting its use for efficient study design provided sufficient coverage is achieved.

ONT LRS represents a powerful tool for comprehensive genome analysis, offering unparalleled advantages for SV detection and the interrogation of complex genomic regions. However, its application for high-accuracy SNV and indel calling requires careful consideration of genomic context and adequate sequencing depth. These findings provide practical guidance for optimising experimental design and analysis strategies, emphasising the complementary roles of LRS and SRS in achieving a complete understanding of genomic variation for both research and potential clinical applications.

## Figures and Tables

**Figure 1 ijms-26-04492-f001:**
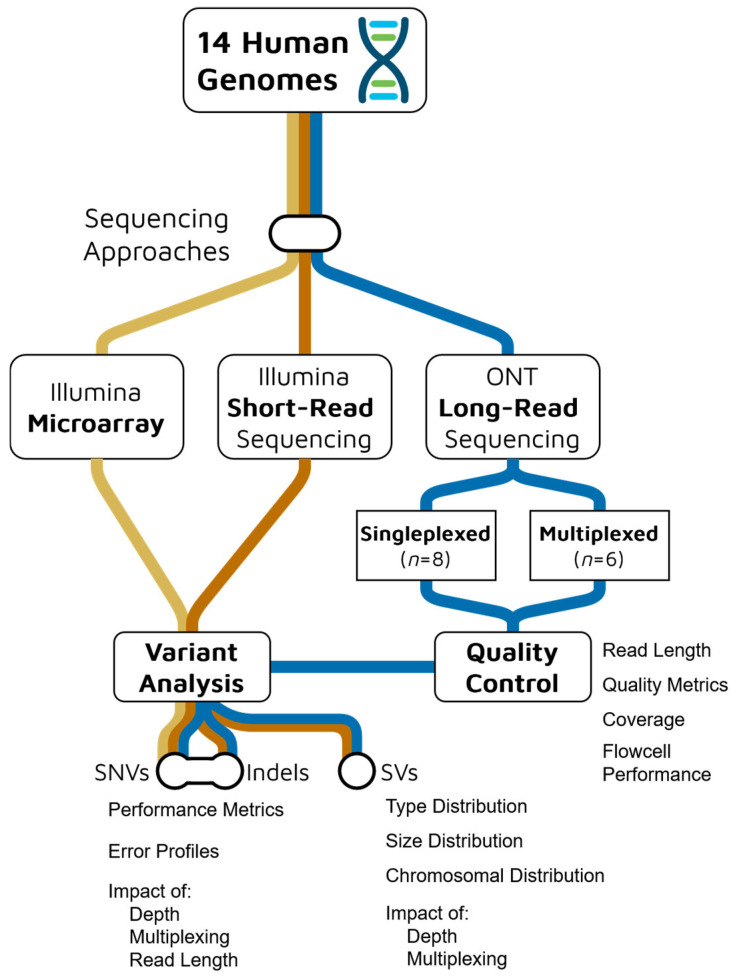
Experimental design and analysis workflow for comparing Oxford Nanopore Technologies (ONT) long-read sequencing (blue) with Illumina short-read sequencing (orange) and microarray (yellow) platforms across 14 human genomes.

**Figure 2 ijms-26-04492-f002:**
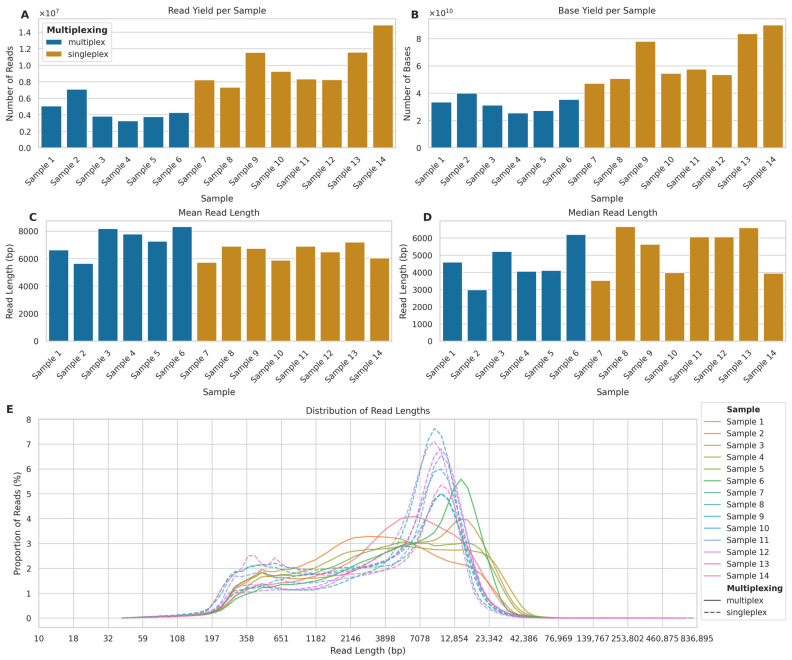
Comparison of read yields and lengths between multiplexed and singleplexed ONT long-read sequencing runs. (**A**) Number of reads per sample (×10^7^) for multiplexed (blue, Samples 1–6) and singleplexed (orange, Samples 7–14) runs. (**B**) Total bases per sample (×10^10^). (**C**) Mean and (**D**) Median read length in base pairs (bp). (**E**) Read length distribution across samples (solid lines: multiplexed; dashed lines: singleplexed) showing proportion of reads (%) in 100 logarithmic bins from 10 bp to 836,895 bp.

**Figure 3 ijms-26-04492-f003:**
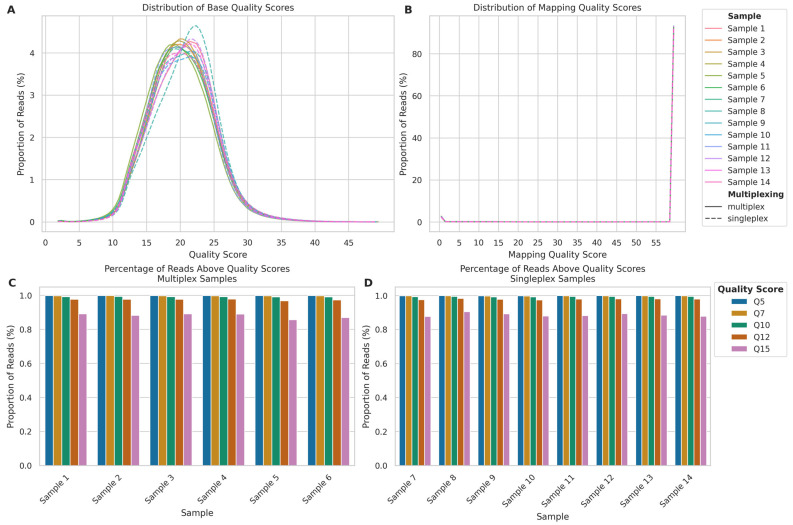
Comparison of sequencing quality metrics between multiplexed and singleplexed ONT long-read sequencing runs. (**A**) Read quality score distributions for multiplexed (Samples 1–6, solid lines) and singleplexed samples (Samples 7–14, dashed lines) calculated by the Dorado basecaller. (**B**) Mapping quality score distributions, calculated by minimap2. (**C**) Percentage of reads above quality thresholds (Q5–Q15) for multiplexed Samples 1–6. (**D**) Percentage of reads above quality thresholds for singleplexed Samples 7–14.

**Figure 4 ijms-26-04492-f004:**
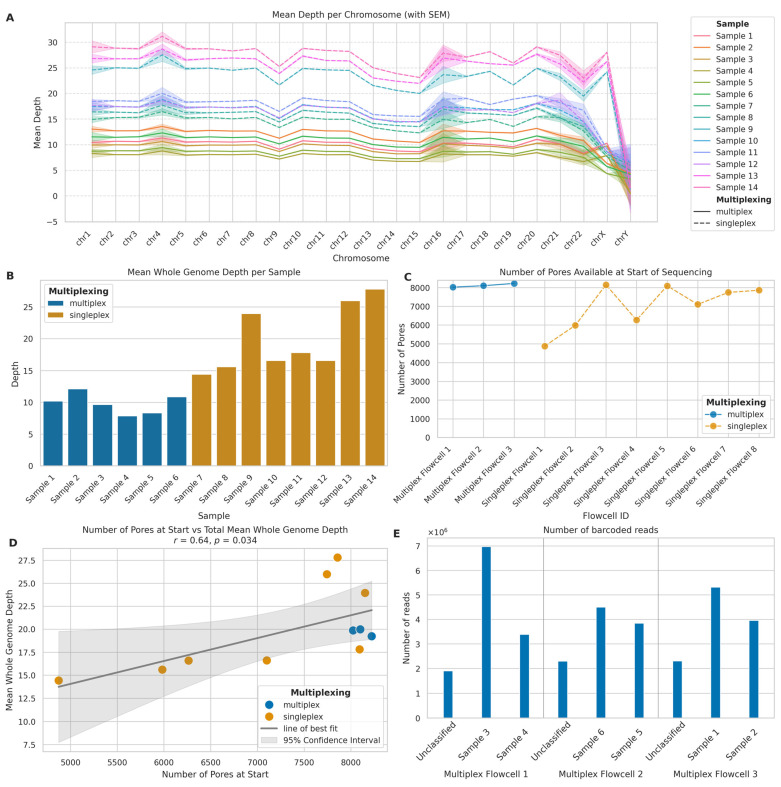
Comparison of multiplexed and singleplexed ONT long-read sequencing performance. (**A**) Mean sequencing depth per chromosome for multiplexed (solid lines) and singleplexed (dashed lines) samples. Shaded areas: standard error of the mean (SEM). (**B**) Mean whole genome sequencing depth per sample for multiplexed (blue) and singleplexed (orange) runs. (**C**) Initial pore availability across flowcells for multiplexed (blue) and singleplexed (orange) runs. (**D**) Correlation between initial pore count and mean whole genome depth. Blue: multiplexed; orange: singleplexed. Solid line: linear regression; shaded area: 95% CI. (**E**) Read distribution across barcoded samples and unclassified reads in multiplexed flowcells.

**Figure 5 ijms-26-04492-f005:**
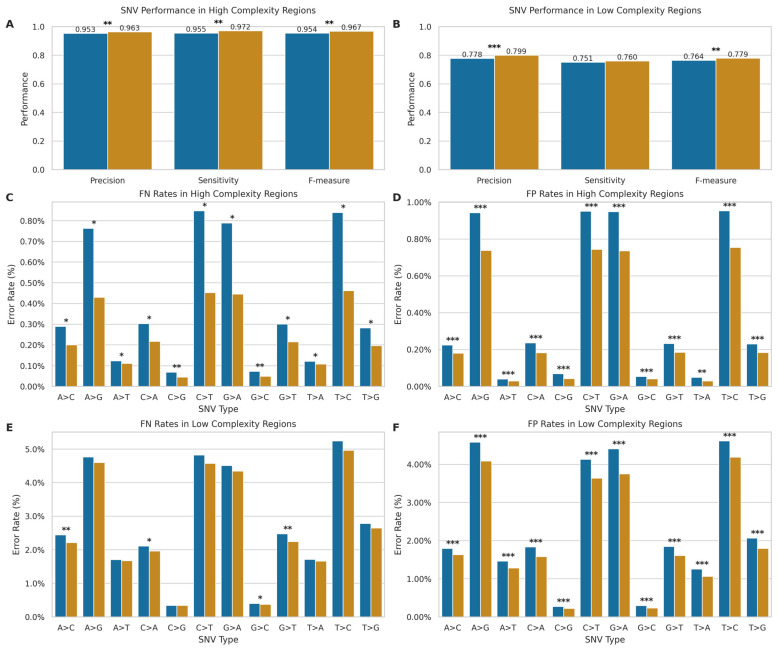
Comparison of SNV detection performance between ONT LRS and SRS. (**A**,**B**) Performance metrics (precision, sensitivity, F-measure) in high- ((**A**); LRS: *n* = 9,563,096; SRS: *n* = 9,476,434) and low-complexity ((**B**); ONT: *n* = 482,788; Illumina: *n* = 473,458) regions. Blue: long-read; orange: short-read. (**C**–**F**) SNV-specific error rates showing false negatives (**C**,**E**) and false positives (**D**,**F**) in high- and low-complexity regions. The *x*-axis indicates nucleotide substitutions. * *p* < 0.05, ** *p* < 0.01, *** *p* < 0.001.

**Figure 6 ijms-26-04492-f006:**
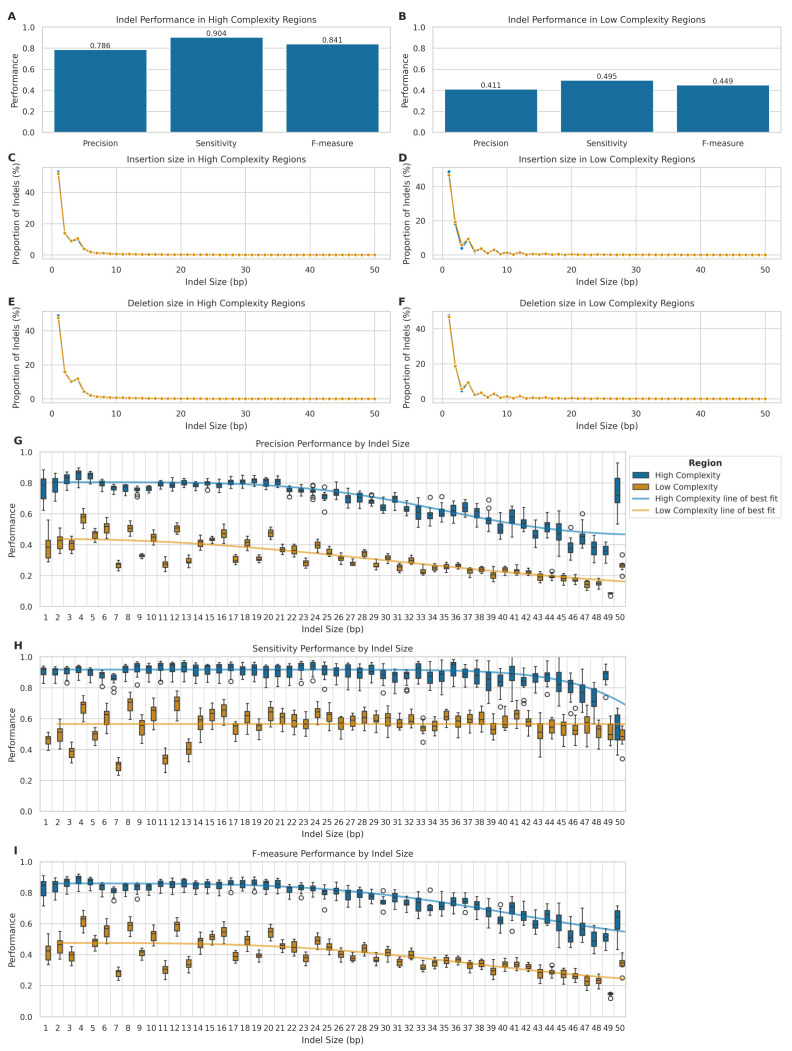
Comparison of indel detection performance between LRS and SRS. (**A**,**B**) Precision, sensitivity, and F-measure for LRS indel detection in high- ((**A**); *n* = 3,706,601) and low-complexity ((**B**); *n* = 15,288,545) regions. (**C**–**F**) Size distribution of indels detected by LRS and SRS: insertions (**C**,**D**) and deletions (**E**,**F**) in high- and low-complexity regions. (**G**–**I**) LRS indel calling metrics by indel length (1–50 bp) in high- (blue) and low-complexity (orange) regions: precision (**G**), sensitivity (**H**), and F-measure (**I**).

**Figure 7 ijms-26-04492-f007:**
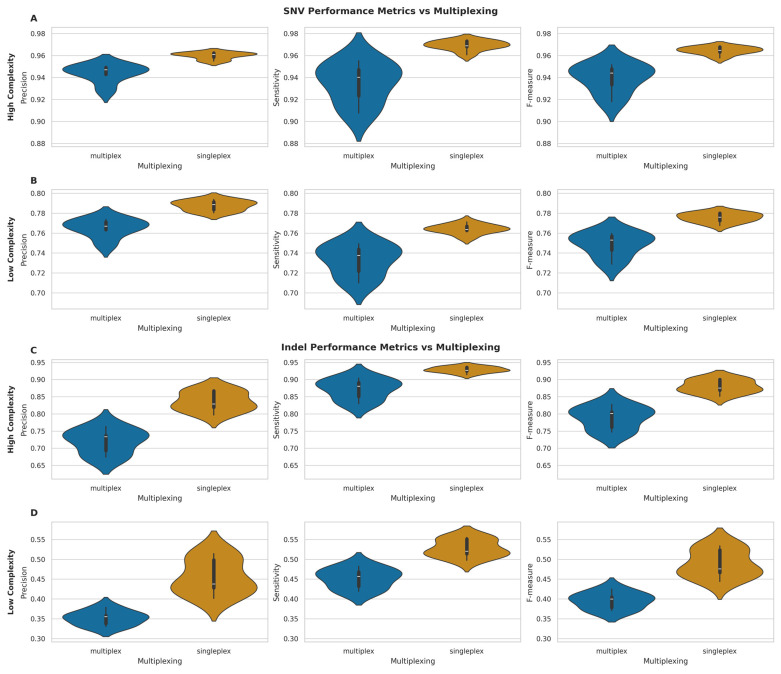
Performance metrics comparison between multiplexed and singleplexed samples for SNV and indel detection. (**A**,**B**) Violin plots of precision, sensitivity, and F-measure for SNVs in high- (**A**) and low-complexity (**B**) genomic regions. (**C**,**D**) Violin plots of same metrics for indels in high- (**C**) and low-complexity (**D**) regions. All plots compare multiplexed (blue) vs. singleplexed (orange) samples, with white bars indicating median values.

**Figure 8 ijms-26-04492-f008:**
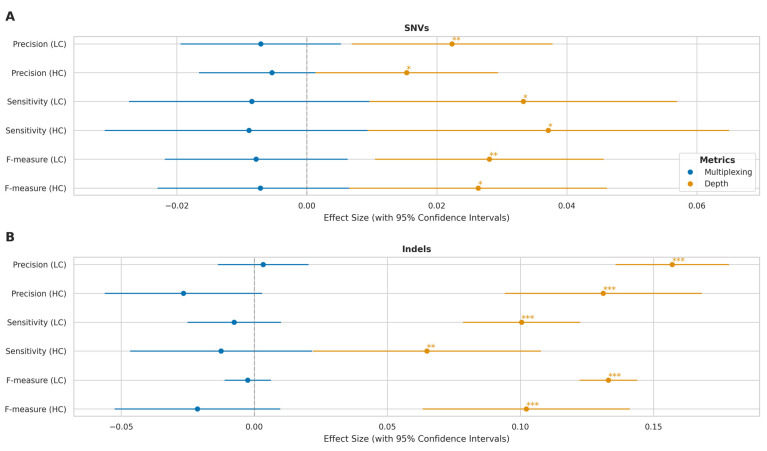
Multiplexing and sequencing depth effects on variant calling across genomic complexity regions. (**A**) Forest plot showing effect sizes with 95% confidence intervals for SNV detection metrics (precision, sensitivity, F-measure) in high-complexity (HC) and low-complexity (LC) regions. Blue: multiplexing effects (relative to singleplex); orange: log-transformed sequencing depth effects. (**B**) Corresponding effect sizes for indel detection metrics. *x*-axis shows magnitude and direction of effects (negative values indicate decreased performance). Statistical significance: * *p* < 0.05, ** *p* < 0.01, *** *p* < 0.001.

**Figure 9 ijms-26-04492-f009:**
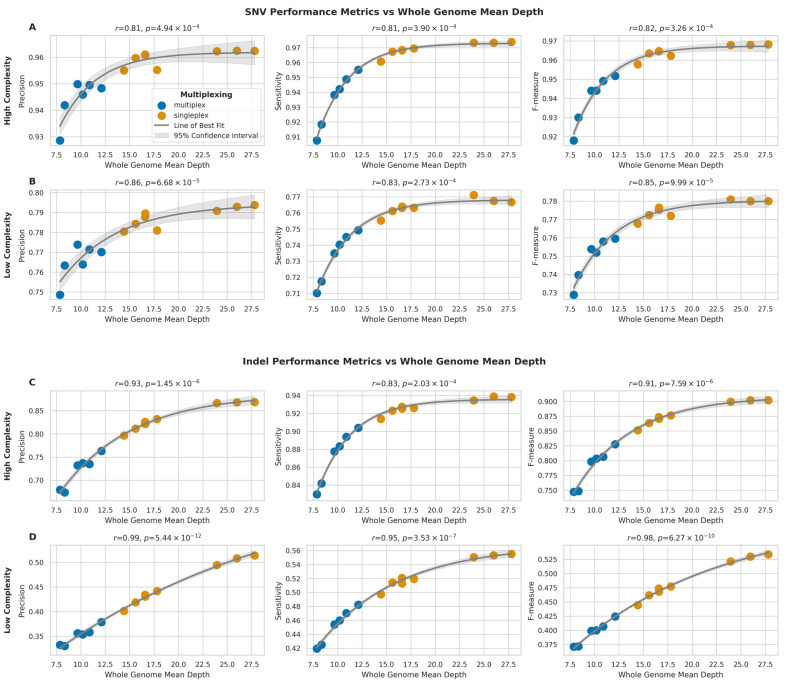
Impact of sequencing depth on SNV and indel detection performance across genomic regions (**A**,**B**) Relationship between whole genome mean depth and performance metrics (precision, sensitivity, F-measure) for SNV detection in high- (**A**) and low-complexity (**B**) regions. (**C**,**D**) Relationship between whole genome mean depth and performance metrics for indel detection in high- (**C**) and low-complexity (**D**) regions. Grey curves represent asymptotic function fits with 95% confidence intervals (light grey). Pearson correlation coefficients (*r*) with *p*-values are shown.

**Figure 10 ijms-26-04492-f010:**
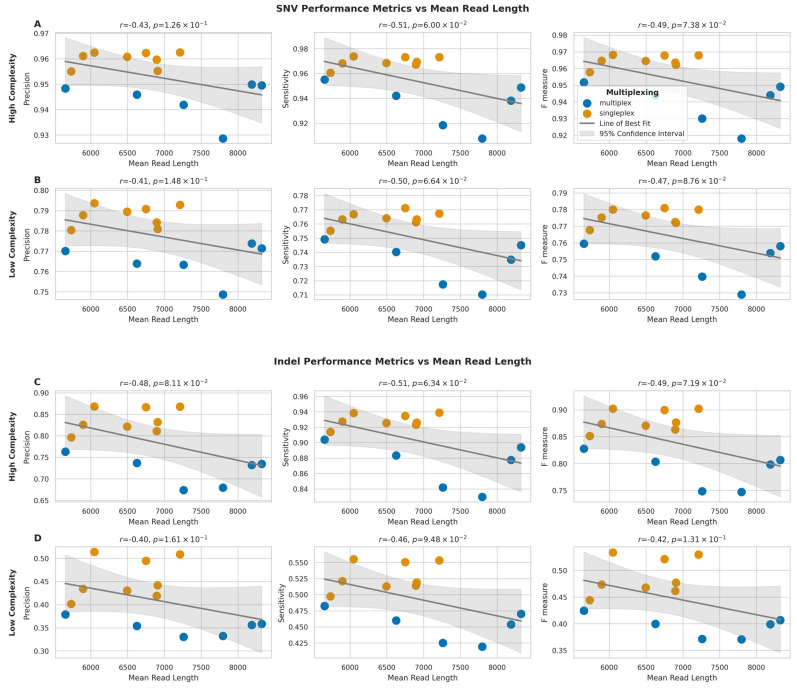
Impact of mean read length on SNV and indel detection performance across genomic regions. (**A**,**B**) Relationship between mean read length and performance metrics (precision, sensitivity, F-measure) for SNV detection in high- (**A**) and low-complexity (**B**) regions. (**C**,**D**) Relationship between mean read length and performance metrics for indel detection in high- (**C**) and low-complexity (**D**) regions. Grey lines show best fit with 95% confidence intervals (shaded). Pearson correlation coefficients (*r*) with *p*-values are shown.

**Figure 11 ijms-26-04492-f011:**
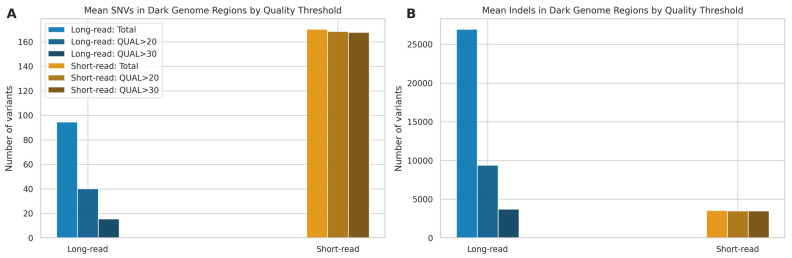
Mean variant counts in dark genome regions by technology and quality threshold. (**A**) Mean SNVs and (**B**) mean indels detected using LRS and SRS. Bars show total mean counts and mean counts remaining after applying quality thresholds (QUAL > 20/30).

**Figure 12 ijms-26-04492-f012:**
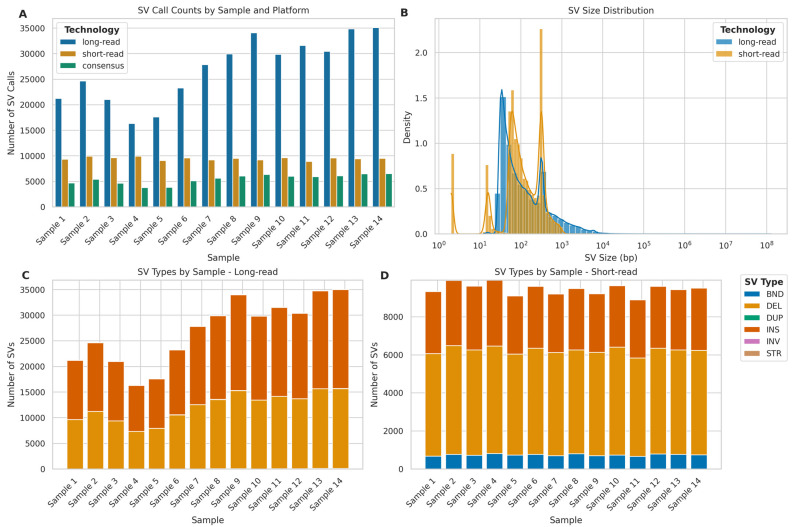
Comparative analysis of structural variant (SV) detection using long-read (ONT) and short-read (Illumina) sequencing platforms. (**A**) SV calls per sample: long-read (blue), short-read (orange), and consensus calls (green). (**B**) SV size distribution (log scale) for long-read (blue) and short-read (orange) platforms. (**C**) Distribution of SV types; BND (breakend), DEL (deletion), DUP (duplication), INS (insertion), INV (inversion), STR (short tandem repeat) detected by long-read sequencing across samples. (**D**) Distribution of SV types detected by short-read sequencing across samples.

**Figure 13 ijms-26-04492-f013:**
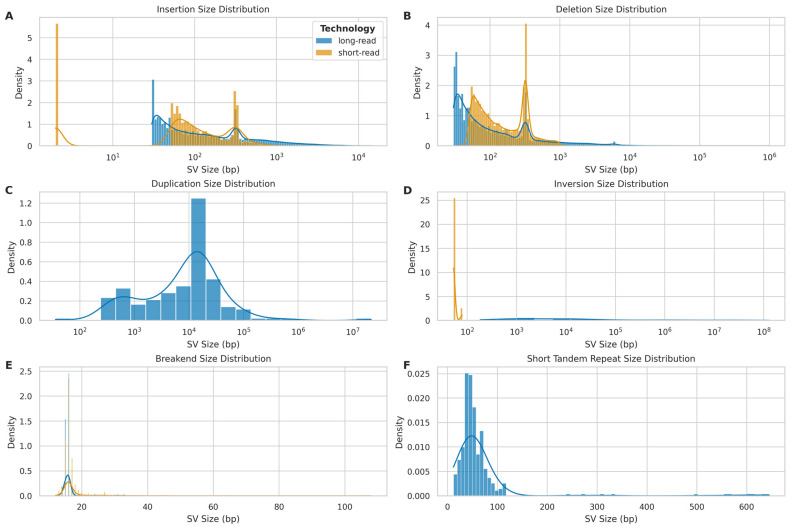
Size distribution of different structural variant (SV) types detected by long-read (ONT) and short-read (Illumina) sequencing platforms. (**A**) Insertion size distribution: long-read (blue) vs. short-read (orange). (**B**) Deletion size distribution: long-read (blue) vs. short-read (orange). (**C**) Duplication size distribution: primarily detected by long-read platform (blue). (**D**) Inversion size distribution: long-read (blue) vs. short-read (orange). (**E**) Breakend (BND) size distribution: long-read (blue) vs. short-read (orange). (**F**) Short tandem repeat (STR) size distribution: exclusively detected by long-read platform (blue).

**Figure 14 ijms-26-04492-f014:**
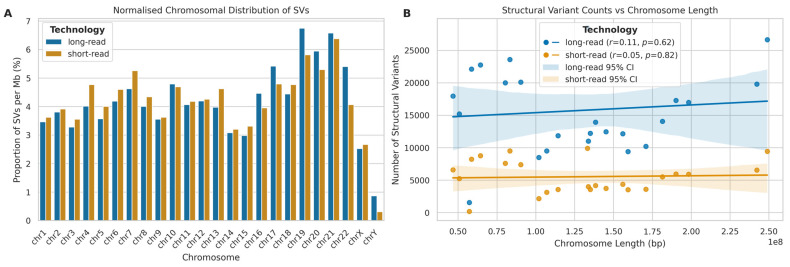
Chromosomal distribution and correlation analysis of SVs detected by long-read (ONT) and short-read (Illumina) sequencing. (**A**) Normalised chromosomal distribution of SVs (% per Mb) for long-read (blue) and short-read (orange) platforms. (**B**) Correlation between SV counts and chromosome length (bp) for both platforms. Lines represent best fit with 95% confidence intervals. Pearson correlation coefficients (r) provided.

**Figure 15 ijms-26-04492-f015:**
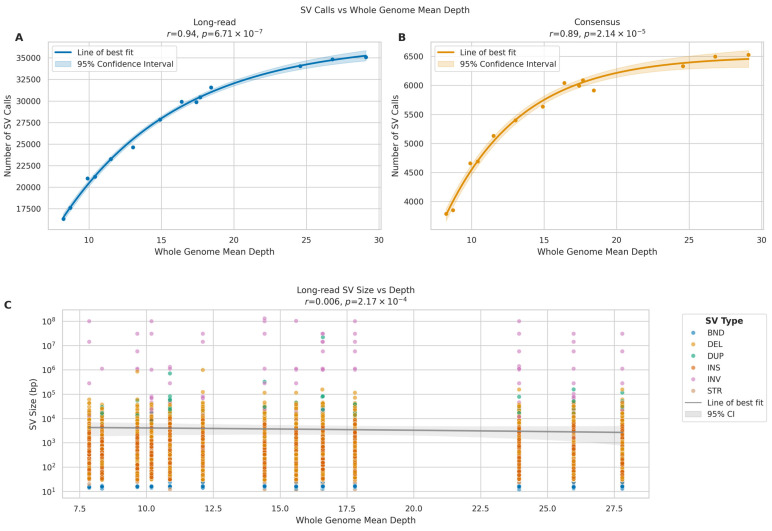
Impact of sequencing depth on SV detection in long-read (ONT). (**A**,**B**) Scatter plots relating whole genome mean depth to SV call numbers for long-read (**A**) and consensus calls (**B**). Lines show best fit with 95% confidence intervals (shaded). Pearson correlation coefficients (r) and *p*-values provided. (**C**) SV size distribution across sequencing depths for long-read data. *Y*-axis shows SV size (log scale). Colours indicate SV types: INS (insertions), DEL (deletions), DUP (duplications), INV (inversions), BND (breakends), and STR (short tandem repeats).

## Data Availability

The datasets presented in this article are not readily available because the data are part of an ongoing study. Requests to access the datasets should be directed to Alfredo Iacoangeli (alfredo.iacoangeli@kcl.ac.uk).

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
