# Peer review of "Investigating the Performance of Oxford Nanopore Long-Read Sequencing with Respect to Illumina Microarrays and Short-Read Sequencing"

_ijms, 2025, doi:10.3390/ijms26104492_

Round 1
Reviewer 1 Report
Comments and Suggestions for Authors
[IJMS] Manuscript ID: ijms-3535881
The MS by Renato et al. compared the two main sequencing methods, Oxford Nanopore long-read sequencing (LRS), Illumina microarrays and short-read sequencing (SRS), and evaluated their performance in variant detection for genomics research. It evaluated the performance for various genetic variants across genomic contexts, and also examined the impact of some key experimental factors, multiplexing, depth, and read length, based on 14 human genomes. Based on the obtained results, a multi-platform benchmark was proposed. By the findings presented in the MS, it may provide valuable insights for optimizing ONT LRS applications in genomic research and clinical diagnostics. Generally, the MS provide some interesting and helpful information for the related academic audience. The MS is well-organized an presented. But, there are still some points need to modify before acceptance.
1.Generally, this length of the MS is somewhat too long (total 35 pages in review version). please consider to reduce the length.
2.For the comparative study of the sequencing methods used in this study, only 14 human genomes was used. Is it enough to perform this study? Please clarify it.
3.Title, please capitalize the first letter the title.
4.Line 7-21, Please delete the hyperlinks for the emails addresses for the authors.
5. Line 38-39, keywords, consider to add, Illumina short-read sequencing
6.Line 43, the introduction part, please consider to simplify the content of this section to reduce the length of this part.
7.Please recheck the format errors in the whole MS, such as L133, the subtitle, 2.1.1. Sequencing Yields.
8.line 531,should 103 (with superscript), also for line 570, “10-4” , Check the others in the MS.
9. Some letters, p, n, R, r2, etc, should be italic. Please recheck.
10. Please recheck the format of the reference part.
11. 1, 9, 16, the author list is too long, please refer the the guide for author of IJMS to adjust the format.

Author Response
We thank the Reviewer for their positive assessment. We appreciate the time and effort dedicated to reviewing our work and providing constructive feedback. We found the comments insightful and believe that addressing them has significantly improved the clarity, focus, and overall quality of our paper.
- Comment: Generally, this length of the MS is somewhat too long (total 35 pages in review version). please consider to reduce the length.
- Response: We acknowledge the reviewer's concern regarding the manuscript length. This study provides a comprehensive benchmark across three platforms (LRS, SRS, microarrays), multiple variant types (SNVs, Indels, SVs), different genomic contexts (high-complexity, low-complexity, dark regions), and assesses the impact of key experimental variables (multiplexing, depth, read length). Presenting these detailed analyses and comparisons inherently requires substantial space. However, we have carefully reviewed the manuscript, particularly the Introduction and Discussion sections (as suggested in point 6), and have condensed the text where possible without compromising the essential details and context required for understanding the study's scope and findings. However, because of the additional analyses requested by other reviewers, the overall length of the manuscript has not changed despite the addition of several new sections. I hope the reviewer appreciates our effort to at least avoid extending the paper further. We believe the current length is justified by the breadth and depth of the benchmarking performed, but we could consider moving some sections to the supplementary if the reviewer requires.
- Comment: For the comparative study of the sequencing methods used in this study, only 14 human genomes was used. Is it enough to perform this study? Please clarify it.
- Response: We thank the reviewer for this question regarding the sample size. While larger cohort studies exist for single platforms, our study's strength lies in the direct, multi-platform comparison (ONT LRS, Illumina SRS, and Illumina Microarrays) performed on the same 14 individuals. These were all the samples available to us for which data from the three platforms was available. Another strength of our study is that for these 14 samples we had whole-genome data which provided millions of variants per sample. For a comparison, the genome in a bottle dataset, only contains whole-genome data from XX samples. Generating and analysing this matched, multi-modal data is resource-intensive. Furthermore, within the LRS data, we generated both singleplexed and multiplexed datasets, allowing for analysis of experimental variables. This design provides considerable data diversity (different platforms, different LRS protocols, varying depths) from these 14 samples, enabling robust comparisons of technological performance across different genomic contexts and experimental conditions. While we acknowledge the sample size as a limitation in the Discussion section (Section 3.7), we believe it is sufficient for the detailed, comparative benchmarking objectives of this specific study, providing valuable insights into relative platform performance.
- Comment: Title, please capitalize the first letter the title.
- Response: Thank you for pointing this out. We have corrected the title capitalisation.
- Comment: Line 7-21, Please delete the hyperlinks for the emails addresses for the authors.
- Response: We apologize for this oversight. The email hyperlinks in the author affiliation section (Lines 7-21) have been removed in the revised manuscript as requested.
- Comment: Line 38-39, keywords, consider to add, Illumina short-read sequencing
- Response: Thank you for the suggestion. We have added "Short-read sequencing" to the keyword list in the revised manuscript.
- Comment: Line 43, the introduction part, please consider to simplify the content of this section to reduce the length of this part.
- Response: Following the reviewer's suggestion and in conjunction with point 1, we have reviewed the Introduction section. We have condensed the background information where feasible, aiming for greater conciseness while still providing the necessary context on the different technologies and the rationale for our comparative study.
- Comment: Please recheck the format errors in the whole MS, such as L133, the subtitle, 2.1.1. Sequencing Yields.
- Response: We apologise for any formatting inconsistencies. We have thoroughly reviewed the entire manuscript for formatting errors, including heading levels (such as the example at the former Line 133, now adjusted), spacing, and overall consistency, to ensure adherence to the journal's style guidelines.
- Comment: line 531,should 103 (with superscript), also for line 570, “10-4” , Check the others in the MS.
- Response: Thank you for catching this formatting issue. We have carefully checked the manuscript and corrected the formatting for scientific notation (e.g., 10³, 10⁻⁴, 10⁸ bp) throughout the text and figures to use proper superscripts for exponents.
- Comment: Some letters, p, n, R, r2, etc, should be italic. Please recheck.
- Response: We agree. We have meticulously reviewed the manuscript and ensured that all standard statistical symbols and variables (e.g., p, n, r, R², SD, CV, IQR) are correctly italicised according to the journal guidelines.
- Comment: Please recheck the format of the reference part.
- Response: We have carefully rechecked the entire reference list to ensure it adheres to the International Journal of Molecular Sciences formatting guidelines.
- Comment: 1, 9, 16, the author list is too long, please refer the the guide for author of IJMS to adjust the format.
- Response: We understand the concern about long author lists in references. We have checked the IJMS guidelines and limited the maximum number of non-abbreviated authors in the reference list to 10.
Reviewer 2 Report
Comments and Suggestions for Authors
Performance of long-read sequencing comparing with short read and microarray technologies.
This is an interesting benchmarking study aiming to compare the performance of long-read sequencing (LRS) vs short-read (SRS) and genotyping in human genomes.
The method section is described in detail and the custom pipeline is available in github.
They compare the impact of sequencing depth and read length on variant calling for the LRS and SRS.
Previous studies have been systematically addressing this issue (Ebbert, 2019). The authors should remark what new information is generated with this study and compare the results with previous studies describing the dark region of the human genome.
The authors should compare the outcome of long-read in the dark region of the genome previously describe, by selecting a random number of dark regions and comparing then with the output of the short read sequencing and the genotyping.
The way that they have presented the results show few differences between short and long read sequencing and a specific analysis of dark regions will show more clearly the differences in the technology.
I also would suggest to add a conclusion paragraph at the end of the discussion section.
Questions
- Will the authors recommend multiplexing for LRS using nanopore sequencing?
- On Page 9, line 257 you state in high complexity regions, SRS overperforms LRS, however, the performance is lower in low complexity regions. Can the authors clarify the rationale for this? Is this a misalignment of the LRS?
Author Response
We thank the Reviewer for their insightful comments. We appreciate the time and effort dedicated to reviewing our work and providing constructive feedback. We believe that addressing them has significantly improved the overall quality of our paper.
- Comment: Previous studies have been systematically addressing this issue (Ebbert, 2019). The authors should remark what new information is generated with this study and compare the results with previous studies describing the dark region of the human genome.
- Response: We thank the reviewer for highlighting the importance of contextualizing our findings within previous work, particularly regarding the "dark genome." Ebbert et al. (2019) provided a crucial definition of these regions based on SRS mapping difficulties. Our revised manuscript now builds upon this by providing a direct, multi-platform benchmark within these challenging regions, comparing ONT LRS against SRS. Our key contributions beyond prior work include:
- We added a new section for analysis of the dark genome (Section 2.7, Figure 11) which explicitly quantifies the significant impact of standard quality filtering on LRS calls in these regions, showing that while LRS can generate many raw indel calls, the yield of high-confidence SNVs/Indels (compared against SRS standards) is low after filtering. This provides a practical perspective on base-level accuracy challenges for LRS in these specific contexts using current standard methods.
- We juxtapose the SNV/Indel findings with the clear advantage of LRS in detecting SVs genome-wide (Section 2.8, Figure 12), many of which reside in or span complex/dark regions, highlighting where LRS truly excels despite base-level accuracy challenges at specific loci within repeats.
- Our findings align with the expectation that LRS improves mappability in dark regions, but we add the insight that achieving high base-level accuracy (for SNVs/Indels) remains challenging, requiring stringent filtering (Section 2.7). This contrasts with LRS's strong performance in SV detection.
We have revised the Discussion (Sections 3.4) to more clearly articulate these points, emphasising the specific contributions of our study relative to prior work like Ebbert et al. and highlighting the practical implications of our findings for variant calling in dark regions.
- Comment: The authors should compare the outcome of long-read in the dark region of the genome previously describe, by selecting a random number of dark regions and comparing then with the output of the short read sequencing and the genotyping. The way that they have presented the results show few differences between short and long read sequencing and a specific analysis of dark regions will show more clearly the differences in the technology.
- Response: We appreciate the reviewer's suggestion to further emphasize the differences between technologies within dark regions. Our revised manuscript now includes Section 2.7 ("Dark Genome Analysis") which systematically analyses all variants detected by LRS and SRS falling within the entire set of dark regions defined by Ebbert et al., rather than a random subset. We believe this comprehensive analysis across all defined dark regions provides a robust picture. It highlights that while LRS can map reads and generate calls in these regions (especially for Indels), translating this into high-confidence base-level variants against SRS remains a challenge.
- Comment: I also would suggest to add a conclusion paragraph at the end of the discussion section.
- Response: Thank you for this suggestion. We have added a dedicated section "4. Conclusions" immediately following the Discussion, as requested.
- Question: Will the authors recommend multiplexing for LRS using nanopore sequencing?
- Response: Our findings provide a nuanced answer. The ANCOVA analysis (Figure 8) indicates that after controlling for sequencing depth, the direct negative impact of multiplexing itself on variant calling accuracy (SNVs and Indels) is minimal and non-significant for all metrics. However, our quality control data (Figure 2A, B; Figure 4B) clearly shows that multiplexed runs (2 samples/flowcell in our case) yielded significantly lower sequencing depth per sample compared to singleplexed runs. Since sequencing depth strongly influences variant calling performance (Figure 9), the key consideration is whether the depth achieved through multiplexing is sufficient for the specific research question.
For applications like SV detection or general genome coverage where moderate depth (e.g., 15-25x) might suffice, multiplexing is a viable and cost-effective strategy.
For applications requiring high accuracy for SNV/indel detection, especially in low-complexity or dark regions, the higher depth typically achieved with singleplexing might be preferable, or higher-yield flowcells/longer runs would be needed if multiplexing. Therefore, we recommend multiplexing cautiously, ensuring that the experimental design achieves the necessary sequencing depth for the intended downstream analysis. This nuance is discussed in Sections 3.1 and 3.5 of the revised manuscript.
- Question: On Page 9, line 257 you state in high complexity regions, SRS overperforms LRS, however, the performance is lower in low complexity regions. Can the authors clarify the rationale for this? Is this a misalignment of the LRS?
- Response: Thank you for asking for clarification on this important point. The difference in relative performance between high and low complexity regions stems from the interplay between basecalling accuracy and read mappability, rather than LRS misalignment.
In high complexity regions, reads from both SRS and LRS generally map accurately and uniquely. The primary determinant of SNV calling performance here is the intrinsic base-calling accuracy of the technology. Illumina SRS has a higher per-base accuracy (~Q30 or 0.1% error rate) compared to ONT LRS (~Q20 modal accuracy or ~1% error rate, though average accuracy is lower). This higher accuracy gives SRS a slight edge in precision and sensitivity for SNVs in well-characterised genomic regions (Figure 5A). LRS's slightly higher base error rate contributes to its lower performance here.
On the other hand, low complexity regions characterized by repeats, homopolymers, etc., which pose significant challenges for read mapping, especially for short reads. Short reads often cannot span repeats, leading to ambiguous mapping (reads mapping to multiple locations) or failed mapping. This significantly reduces SRS's ability to confidently call variants (lower sensitivity, Figure 5B) and can also lead to mapping errors causing false positives (lower precision, Figure 5B).
Long reads have a much higher chance of spanning these repetitive regions entirely, allowing for more confident and unique mapping. This mapping advantage helps LRS maintain its sensitivity relatively better than SRS in low-complexity regions (the sensitivity gap closes, Figure 5B). However, LRS's base-calling errors still persist, which means its precision remains slightly lower than SRS's precision (derived from the fewer, but more accurately base-called, SRS reads that do map confidently).
It's not that LRS misalignment increases in low-complexity regions (it likely decreases relative to SRS), but rather that SRS's mapping failures become the dominant limitation.
Round 2
Reviewer 2 Report
Comments and Suggestions for Authors
No further comments.